# Robust Bayesian Recourse

**Tuan-Duy H. Nguyen**[1]     **Ngoc Bui**[1]     **Duy Nguyen**[1]     **Man-Chung Yue**[2]     **Viet Anh Nguyen**[1]

[1]VinAI Research, Vietnam
[2]The University of Hong Kong

## Abstract

Algorithmic recourse aims to recommend an informative feedback to overturn an unfavorable machine learning decision. We introduce in this paper the Bayesian recourse, a model-agnostic recourse that minimizes the posterior probability odds ratio. Further, we present its min-max robust counterpart with the goal of hedging against future changes in the machine learning model parameters. The robust counterpart explicitly takes into account possible perturbations of the data in a Gaussian mixture ambiguity set prescribed using the optimal transport (Wasserstein) distance. We show that the resulting worst-case objective function can be decomposed into solving a series of two-dimensional optimization subproblems, and the min-max recourse finding problem is thus amenable to a gradient descent algorithm. Contrary to existing methods for generating robust recourses, the robust Bayesian recourse does not require a linear approximation step. The numerical experiment demonstrates the effectiveness of our proposed robust Bayesian recourse facing model shifts. Our code is available at `https://github.com/VinAIResearch/robust-bayesian-recourse`.

## 1  INTRODUCTION

Human constantly embark on multiple temporally-extended planning problems throughout the course of their lifespan, and we have several layers of means-end planning in order to achieve the desired goals. For example, to have a successful career as a machine learning researcher, an individual needs to put in persistent effort from their early education to their post-graduate studies, which may span over the course of over twenty years with numerous significant milestones to achieve. Two of these important milestones are the PhD admission and the job application, and arguably, a favorable outcome at these two milestones may propel an individual's career on a more auspicious trajectory than a negative outcome. To aid the committee to make better decisions, machine learning models are increasingly used in both university admission [Waters and Miikkulainen, 2014] and job hiring [Sajjadiani et al., 2019]. A similar trend takes place in credit loan applications [Siddiqi, 2012], healthcare [Mertes et al., 2021] and many others.

The increasing reliance on and the long impact of algorithmic decisions raise significant requirements on the trustworthiness and explainability of the machine learning models. These requirements become more urgent as black-box, complex models are also gaining spotlight attraction due to their superior performance [Garisch and Merchant, 2019]. Post-hoc explanations, which extracts human-understandable explanations, may benefit individuals to understand machine-produced decisions [Kenny et al., 2021]. A post-hoc method must demonstrate why unfavorable predictions are made, and possibly how an input would have been to obtain a favorable predicted outcome. If the inputs encode the characteristics of human individuals, then a possible post-hoc explanation may come in the form of a recourse. A recourse recommends the actions that an individual should take in order to receive an alternate algorithmic outcome [Ustun et al., 2019]. Consider an applicant who is rejected for a particular job, a recourse may come in the form of personalized recommendations such as "complete a 6-month full-stack engineer internship" or "score 20 more points in the ability test", along with the promise that if the applicant successfully implement the necessary action then the algorithm will return a favorable outcome.

Several approaches has been proposed to provide recourses for machine learning models [Karimi et al., 2021, Stepin et al., 2021, Mishra et al., 2021, Artelt and Hammer, 2019, Pawelczyk et al., 2021]. Wachter et al. [2018] used a gradient-based approach to find nearest counterfactual to the original instance. Ustun et al. [2019] proposed an integer

*Accepted for the 38th Conference on Uncertainty in Artificial Intelligence* (UAI 2022).

programming approach to generate actionable recourses for linear classifiers. Karimi et al. [2020] proposed a model-agnostic approach to generate nearest counterfactual explanations while Poyiadzi et al. [2020] generates counterfactuals that are actionable and supported by the "feasible paths" of actions. Pawelczyk et al. [2020] find a counterfactual explanation with an upper bound for the costs of counterfactual explanations under predictive multiplicity. Mothilal et al. [2020] proposed a framework for generating and evaluating a diverse set of counterfactual explanations based on determinantal point processes. Bui et al. [2022] proposed an uncertainty quantification tool to compute the bounds of the probability of validity of a set of counterfactual explanations and enhanced the validity of this set via a correction tool.

These aforementioned approaches all assume that the underlying machine learning models do not change over time. In practice, this assumption is easily violated as experts update the machine learning system frequently due to data distribution shifts [Quionero-Candela et al., 2009, Y et al., 2019]. As such, an individual may have accomplished all the recommended actions but the next time they apply for the job, the parameters of the model may already change and the updated model may still recommend a negative outcome. In that case, the recourse becomes useless: it is ineffective in overturning a negative prediction, it incurs cost to the applicant, and at the same time it raises substantial doubts about the recourse [Rawal et al., 2020]. Following this line, a recourse is considered to be robust if it is effective at reversing the algorithmic outcome even under model shifts.

To construct a robust recourse, Upadhyay et al. [2021] proposed ROAR, a framework that leverages adversarial training to hedge against the perturbation of the model parameter. ROAR considers only linear classifiers; for *non*linear classifiers, ROAR first generates a locally linear approximation of the underlying model (e.g., by using LIME [Ribeiro et al., 2016]), then applies the adversarial training procedure with respect to this locally linear surrogate. However, there are multiple downsides when a locally linear model is used to approximate the nonlinear classifier. Recent works have shown that the locally linear model of LIME has some limitations with both its fidelity and robustness. LIME may not be faithful to the underlying model since it might be influenced by input features at a global scale rather than a local scale [White and Garcez, 2019, Laugel et al., 2018]. At the same time, several works [Alvarez-Melis and Jaakkola, 2018, Slack et al., 2020, Agarwal et al., 2021] point out that the explanations generated by LIME and other explanation methods may change significantly for nearby original inputs. Moreover, these explanations are even sensitive to the sampling distribution, and the can deliver different explanations of the same input in different simulation runs. Finally, a recourse which is robust for the linear approximation model may not necessarily be robust respective to the original nonlinear model.

**Contributions.** The goal of this paper is to formulate a model-agnostic recourse, which is also valid subject to potential future shifts of the machine learning models. Compared to existing methods such as ROAR [Upadhyay et al., 2021], our method does not depend on the linear surrogate of the *non*linear predictive model. Instead, our method looks directly into the sampled data points, and employs a Bayesian approach to generate recourses. Potential shifts of the predictive models are engendered by "perturbing" these data samples in an adversarial manner. We contribute concretely the followings.

- In Section 2, we propose the notion of a Bayesian recourse, which minimizes the odds ratio between the posterior probability of negative and positive predicted outcomes. In a non-parametric setting, the likelihood can be approximated using a kernel density estimator built around the data sampled in the neighborhood of the boundary point. This results in the KDE-Bayesian recourse, which can be found by (projected) gradient descent.

- In Section 3, we propose the robust counterpart of the Bayesian recourse problem. This robustification involves smoothing the samples by an isotropic Gaussian convolution, then solving a min-max optimization problem over a Wasserstein-Gaussian mixture conditional ambiguity set. Section 4 details our method of using the optimal transport to form the ambiguity sets on the space of Gaussian mixtures.

- In Section 5, we show that the robust Bayesian recourse problem is amenable to separability and dimensionality reduction, thus the recourse can be constructed efficiently even in high dimensions. Section 6 demonstrates that our recourse also performs competitively on both synthetic and real datasets.

**Notations.** We use $\delta_s$ to denote a Dirac measure supported on point $s$. The space of $p$-by-$p$ symmetric, positive semidefinite matrix is denoted by $\mathbb{S}_+^p$.

## 2 BAYESIAN RECOURSE

We consider a generic covariate $X \in \mathcal{X} = \mathbb{R}^p$ and a binary predicted label $\hat{Y} \in \mathcal{Y} = \{0, 1\}$, where class 0 denotes an *un*favorable outcome while class 1 denotes a favorable one. Given a pre-specified black-box classifier $\mathcal{C}$ and an input $x_0$ with unfavorable prediction, i.e., $\mathcal{C}(x_0) = 0$, the goal of algorithmic recourse is to devise an alternative $x'$ in the vicinity of $x_0$ that satisfies $\mathcal{C}(x') = 1$. The Bayesian recourse imposes a probabilistic viewpoint into this problem: the goal of Bayesian recourse is to devise an alternative in the vicinity of $x_0$ that has high *favorable posterior probability*. In technical terms, consider the joint random vector of covariate-label $(X, \hat{Y}) \in \mathcal{X} \times \mathcal{Y}$, then the class posterior probability of any input $x$ can be represented by the

conditional random variable $\hat{Y}|X = x$.[1]

**Definition 2.1** (Bayesian recourse). *Given an input $x_0$, let $\mathbb{X}$ be a neighborhood around $x_0$. A Bayesian recourse $x_{\mathrm{Bayes}} \in \mathbb{X}$ is an alternative that minimizes the Bayesian posterior odds ratio, i.e.,*

$$x_{\mathrm{Bayes}} \triangleq \arg\min_{x \in \mathbb{X}} \frac{\mathbb{P}(\hat{Y} = 0|X = x)}{\mathbb{P}(\hat{Y} = 1|X = x)},$$

*for some joint distribution $\mathbb{P}$ of $(X, \hat{Y})$ induced by the sampling of the synthetic covariate $X$ and the synthetic predicted label $\hat{Y} = \mathcal{C}(X)$.*

The ratio $\mathbb{P}(\hat{Y} = 0|X = x)/\mathbb{P}(\hat{Y} = 1|X = x)$ is a well-known quantity in Bayesian classification. The posterior probability odds is also a popular ratio in Bayesian statistics, and it has been applied for comparing regression hypotheses Zellner [1981], econometric models Geweke [1994], asset pricing theories McCulloch and Rossi [1991] and collaborative evaluations Hicks et al. [2018].

As $x_{\mathrm{Bayes}}$ minimizes the Bayesian posterior odds ratio, we can argue that $\mathbb{P}(\hat{Y} = 0|X = x_{\mathrm{Bayes}})$ tends to be low, while $\mathbb{P}(\hat{Y} = 1|X = x_{\mathrm{Bayes}})$ tends to be high. We next describe how we can solve the optimization problem to get $x_{\mathrm{Bayes}}$. Note that the posterior probability can be calculated using the Bayes' theorem [Schervish, 1995, Theorem 1.31], and we can instead solve the fractional optimization problem

$$\min_{x \in \mathbb{X}} \frac{\mathbb{P}(\hat{Y} = 0)\mathbb{P}(X = x|\hat{Y} = 0)}{\mathbb{P}(\hat{Y} = 1)\mathbb{P}(X = x|\hat{Y} = 1)}.$$

It is now clear that to find $x_{\mathrm{Bayes}}$, we need the marginal probability of $\hat{Y}$ and the likelihood of $X|\hat{Y}$. Suppose that we can use a sampling mechanism to sample $n$ covariates $\widehat{x}_i$, then query the given classifier to obtain the predicted labels $\widehat{y}_i = \mathcal{C}(\widehat{x}_i)$ to form $n$ pairs $(\widehat{x}_i, \widehat{y}_i)$, $i = 1, \dots, n$. Using these synthetic, labelled samples, we now can formulate the empirical version of Bayesian recourse problem. Let $\mathcal{I}_y = \{i \in [n] : \widehat{y}_i = y\}$ be the indices of samples in class $y \in \mathcal{Y}$. Let $N_y = |\mathcal{I}_y|$ be the number of training samples with class $y$, then we can use $\gamma_y = N_y/n$, the empirical proportion of data for class $y$, as an estimate of $\mathbb{P}(\hat{Y} = y)$.

Next, we take the nonparametric approach to estimate the likelihood $\mathbb{P}(X = x|\hat{Y} = y)$ using a kernel density estimator [Tsybakov, 2008, Section 1]. As a concrete example, we choose the Gaussian kernel with bandwidth $h > 0$, thus the kernel density estimate of the quantity $\mathbb{P}(X = x|\hat{Y} = y)$ is

$$L_{\mathrm{KDE}}(x|\hat{Y} = y) = \frac{1}{N_y} \sum_{i \in \mathcal{I}_y} \exp\left(-\frac{1}{2h^2}\|x - \widehat{x}_i\|_2^2\right).$$

---

[1] In algorithmic recourse, the random variable of interest is the predicted label $\hat{Y}$ induced by the classifier $\mathcal{C}$, not the true label $Y$ of the data-generating process. It is important to keep in mind that the (robust) Bayesian recourse is formulated with respect to the predicted label $\hat{Y}$.

Thus, the empirical version of the Bayesian recourse, termed the KDE-Bayesian recourse, can be found by solving

$$\min_{x \in \mathbb{X}} \frac{\gamma_0 \times L_{\mathrm{KDE}}(x|\hat{Y} = 0)}{\gamma_1 \times L_{\mathrm{KDE}}(x|\hat{Y} = 1)}. \qquad (1)$$

This problem further simplifies to

$$\min_{x \in \mathbb{X}} \frac{\sum_{i \in \mathcal{I}_0} \exp\left(-\frac{1}{2h^2}\|x - \widehat{x}_i\|_2^2\right)}{\sum_{i \in \mathcal{I}_1} \exp\left(-\frac{1}{2h^2}\|x - \widehat{x}_i\|_2^2\right)}$$

by exploiting the definition of $L_{\mathrm{KDE}}$ and $\gamma_y$. In this form, a (projected) gradient descent algorithm can be employed to find the KDE-Bayesian recourse.

There remain two elements to be specified about the formulation of the Bayesian recourse: the sampling scheme to generate covariates $\widehat{x}_i$ and the feasible set $\mathbb{X}$. We discuss these components in the remainder of this section.

**Sampling scheme.** The goal of the sampling scheme is to synthesize covariate data $\widehat{x}_i$ around the boundary to obtain *local* information from the black-box classifier. Toward this goal, we use a local sampling method, similar to Vlassopoulos et al. [2020] and Laugel et al. [2018] as follows.

- Given an instance $x_0$, we choose $K$ nearest counterfactuals $x_1, \dots, x_K$ from the training data that have favorable predicted outcome, that is, $\mathcal{C}(x_k) = 1$ for $k = 1, \dots, K$.

- For each counterfactual $x_k$, we perform a line search to find a point $x_k^b$ that is on the decision boundary and on the line segment joining $x_0$ and $x_k$.

- Among these points $x_k^b$, we choose the nearest point to $x_0$ by setting $x^b \triangleq \arg\min_{x_i^b}\{c(x_i^b, x_0)\}$, where $c(\cdot)$ is the cost function. We then sample $\widehat{x}_i$ uniformly in a neighborhood determined by an $\ell_2$-ball with radius $r_p$ centered on $x^b$.

**Feasible set $\mathbb{X}$.** It is desirable to constrain the recourse in a *strict* neighborhood of distance $\delta$ from the input Venkatasubramanian and Alfano [2020]. Thus, we can impose a feasible set of the form

$$\mathbb{X} = \{x \in \mathcal{X} \ : \ \varphi(x, x_0) \leq \delta\},$$

where $\varphi$ is a measure of dissimilarity on the covariate space $\mathcal{X}$. Alternatively, if we use a boundary sampler as previously discussed, we may also opt for the constraint $\varphi(x, x^b) \leq \delta'$ around the boundary point $x^b$. A good choice of $\varphi$ is the $\ell_1$ distance, which promotes sparse modifications to the input.

In order to construct plausible and meaningful recourses, we could additionally consider the actionability constraints that forbid unrealistic recourses. For example, the gender or race of a person should be considered immutable. Likewise, recourse should not suggest an individual reduce their age to achieve a favorable outcome. These constraints could be easily injected into the definition of the feasible set $\mathbb{X}$, similar

to Upadhyay et al. [2021]. Finding the optimal actionable recourse restricted to this feasible set could be addressed effectively by a projected gradient descent algorithm [Mothilal et al., 2020, Upadhyay et al., 2021].

# 3 ROBUST BAYESIAN RECOURSE

The Bayesian recourse in Definition 2.1 depends on the classifier $\mathcal{C}$ as we query $\mathcal{C}$ to label the samples $\widehat{x}_i$ via $\widehat{y}_i = \mathcal{C}(\widehat{x}_i)$. Thus, inherently, the recourse would possess high posterior probability of favorable outcome with respect to the *present* classifier $\mathcal{C}$. Because the parameters defining $\mathcal{C}$ may be updated, the Bayesian recourse does not guarantee a high probability of favorable outcome with respect to the *future* classifier $\tilde{\mathcal{C}}$. Devising a recourse that has a high probability of future favorable outcome encounters two critical difficulties: first, the classifiers $\mathcal{C}$ and $\tilde{\mathcal{C}}$ are possibly nonlinear, and second, it is nontrivial to predict the shifts in the parameters of $\tilde{\mathcal{C}}$ from the present model $\mathcal{C}$. Existing robust recourse methods such as ROAR [Upadhyay et al., 2021] need to approximate a nonlinear model by a linear model using LIME [Ribeiro et al., 2016], then robustness is represented by perturbations of the parameters of the linear surrogate.

The robust Bayesian recourse takes a completely different path to ensure robustness by removing the need for an intermediate linear surrogate model. The robust Bayesian recourse aims to perturb directly the empirical conditional distributions of $X|\hat{Y} = y$, which then reshapes the decision boundary in the covariate space in an adversarial manner. Holistically, our approach can be decomposed into the following steps:

1. Forming the empirical conditional distributions of $X|\hat{Y} = y$, then smoothen them by convoluting an isotropic Gaussian noise to each data point.

2. Formulating the ambiguity set for each conditional distributions of $X|\hat{Y} = y$.

3. Solving a min-max problem to find the recourse that minimizes the worst-case Bayesian posterior odds ratio.

We now dive into the technical specifications of the robust Bayesian recourse. Remind that the sampling procedure equips us with the samples $(\widehat{x}_i, \widehat{y}_i)_{i=1,\dots,n}$, and $\mathcal{I}_y$ are indices of samples with predicted label $y$. Let $\widehat{\mathbb{P}}_y^\sigma = N_y^{-1} \sum_{i \in \mathcal{I}_y} \delta_{\widehat{x}_i} * \mathcal{N}(0, \sigma^2 I)$ be the *smoothed* empirical conditional distribution of $X|Y = y$, in which $*$ denotes the convolution. Notice that $\widehat{\mathbb{P}}_y^\sigma$ is a mixture of Gaussian with $N_y$ components located at the covariate $\widehat{x}_i$ with isotropic variance $\sigma^2 I$. Smoothing the empirical distribution by convoluting a noise to each sample is also attracting attention recently thanks to its possibility to quantify and enhance the robustness of machine learning models Cohen et al. [2019].

We assume now that the conditional distribution can be perturbed in an ambiguity set $\mathbb{B}_{\varepsilon_y}(\widehat{\mathbb{P}}_y^\sigma)$. This set $\mathbb{B}_{\varepsilon_y}(\widehat{\mathbb{P}}_y^\sigma)$ is

defined as a neighborhood of radius $\varepsilon_y \geq 0$ centered at the nominal distribution $\widehat{\mathbb{P}}_y^\sigma$. The robust Bayesian recourse is defined as the optimal solution of the following problem

$$\min_{x \in \mathbb{X}} \max_{\mathbb{Q}_0 \in \mathbb{B}_{\varepsilon_0}(\widehat{\mathbb{P}}_0^\sigma), \mathbb{Q}_1 \in \mathbb{B}_{\varepsilon_1}(\widehat{\mathbb{P}}_1^\sigma)} \frac{\gamma_0 \mathbb{Q}_0(X = x)}{\gamma_1 \mathbb{Q}_1(X = x)}. \quad (2)$$

Notice that $\mathbb{Q}_y$ is a *conditional* probability measure of $X$ given $\hat{Y} = y$, and thus it is a measure supported on $\mathbb{R}^p$. The value $\mathbb{Q}_y(X = x)$ is also the likelihood of $x$ under the conditional measure $\mathbb{Q}_y$, thus problem (2) can be view as a robust likelihood ratio minimization problem. Here, robustness is defined with respect to the conditional sets $\mathbb{B}_{\varepsilon_y}(\widehat{\mathbb{P}}_y^\sigma)$ in the specific sense: the optimal value of problem (2) constitutes a uniform upper bound of the likelihood ratio over all possible choices of conditional distributions in the sets $\mathbb{B}_{\varepsilon_y}(\widehat{\mathbb{P}}_y^\sigma)$. Further, we have explicitly used $\gamma_y$ as an estimator of the marginal distribution of $\hat{Y}$ in problem (2).

There is an intimate relationship between the KDE-Bayesian recourse problem (1) and the robust Bayesian recourse problem (2). This relationship is established thanks to the smoothing of the empirical conditional distributions, and is highlighted in the following remark.

**Remark 3.1** (Recovery of the KDE-Bayesian recourse). *The smoothed conditional distribution $\widehat{\mathbb{P}}_y^\sigma$ is a mixture of Gaussians, and the likelihood of $x$ under $\widehat{\mathbb{P}}_y^\sigma$ is*

$$\frac{1}{N_y (2\pi)^{\frac{p}{2}} \sigma^p} \sum_{i \in \mathcal{I}_y} \exp\left( -\frac{1}{2\sigma^2} \|x - \widehat{x}_i\|_2^2 \right).$$

*As a consequence, if the ambiguity sets $\mathbb{B}_{\varepsilon_y}(\widehat{\mathbb{P}}_y^\sigma)$ collapse into singletons, that is, $\mathbb{B}_{\varepsilon_y}(\widehat{\mathbb{P}}_y^\sigma) = \{\widehat{\mathbb{P}}_y^\sigma\}$, then problem (2) coincides with the KDE-Bayesian recourse problem (1). Thus, problem (2) can be considered as a robustification of the KDE-Bayesian recourse formulation.*

# 4 WASSERSTEIN-GAUSSIAN MIXTURE AMBIGUITY SETS

The central notion underlying the robust Bayesian recourse problem (2) is the set of probability measures for the covariate $X$ conditional that $Y = y$. A suitable design of the ambiguity set $\mathbb{B}_{\varepsilon_y}(\widehat{\mathbb{P}}_y^\sigma)$ is critical to enable an efficient resolution of problem (2). We here propose a novel design of the ambiguity set by merging ideas from the theory of optimal transport and Gaussian mixtures.

Note that any Gaussian distribution is fully characterized by its mean vector and its covariance matrix. As the smoothed measure $\widehat{\mathbb{P}}_y^\sigma$ is a Gaussian mixture, it is associated with the discrete distribution $\widehat{\nu}_y = N_y^{-1} \sum_{i \in \mathcal{I}_y} \delta_{(\widehat{x}_i, \sigma^2 I)}$ on the space of mean vector and covariance matrix $\mathbb{R}^p \times \mathbb{S}_+^p$.[2]

---

[2] Associated with any mixture of Gaussians $\mathbb{Q}_y$ on $\mathbb{R}^p$ is a

Moreover, define the set

$$\mathbb{S}^p_{\geq \sigma} \triangleq \{\Sigma \in \mathbb{S}^p_+ : \Sigma \succeq \sigma^2 I\} \subset \mathbb{S}^p_+$$

of covariance matrices whose eigenvalues are lower bounded by $\sigma^2 > 0$, where $\sigma^2$ is the isotropic variance of the smoothing convolution. Notice that we explicitly constrain the covariance matrices to be invertible so that the likelihood function of each Gaussian component is well-defined. For any $y \in \{0, 1\}$, we formally define the ambiguity set as

$$\mathbb{B}_{\varepsilon_y}(\widehat{\mathbb{P}}^\sigma_y) \triangleq$$
$$\left\{ \mathbb{Q}_y : \begin{array}{c} \nu_y \in \mathcal{P}(\mathbb{R}^p \times \mathbb{S}^p_{\geq \sigma}),\ \mathbb{W}_c(\nu_y, \widehat{\nu}_y) \leq \varepsilon_y \\ \mathbb{Q}_y \text{ is a Gaussian mixture associated with } \nu_y \end{array} \right\}.$$

Here, $\mathcal{P}(\mathbb{R}^p \times \mathbb{S}^p_{\geq \sigma})$ denotes the set of all possible distributions supported on $\mathbb{R}^p \times \mathbb{S}^p_{\geq \sigma}$. Intuitively, $\mathbb{B}_{\varepsilon_y}(\widehat{\mathbb{P}}^\sigma_y)$ contains all Guassian mixtures $\mathbb{Q}_y$ associated with some $\nu_y$ having a distance less than or equal to $\varepsilon_y$ from the nominal measure $\widehat{\nu}_y$. Thus each measure $\mathbb{Q}_y$ of the random vector $X|Y = y$ is a Gaussian mixture. Each distribution $\nu_y$ is a measure on the space of mean vector-covariance matrix $\mathbb{R}^p \times \mathbb{S}^p_{\geq \sigma}$, and the distance between $\nu_y$ and $\widehat{\nu}_y$ is measured by an optimal transport distance $\mathbb{W}_c$. We will use in this paper the type-$\infty$ Wasserstein distance, which is defined as follows.

**Definition 4.1** (Type-$\infty$ Wasserstein distance). *Let $c$ be a nonnegative, symmetric and continuous ground transport cost on $\Xi \triangleq \mathbb{R}^p \times \mathbb{S}^p_{\geq \sigma}$. The type-$\infty$ Wasserstein distance between two distributions $\nu_1,\ \nu_2 \in \mathcal{P}(\Xi)$ amounts to*

$$\mathbb{W}_c(\nu_1, \nu_2)$$
$$\triangleq \inf_{\lambda \in \Lambda(\nu_1, \nu_2)} \left\{ \operatorname*{ess\,sup}_{\lambda} \left\{ c(\xi_1, \xi_2) : (\xi_1, \xi_2) \in \Xi \times \Xi \right\} \right\},$$

*where $\Lambda(\nu_1, \nu_2)$ is the set of all couplings of $\nu_1$ and $\nu_2$.*

It remains to specify the ground metric $c$ on the space $\mathbb{R}^p \times \mathbb{S}^p_{\geq \sigma}$. Because the space $\mathbb{R}^p \times \mathbb{S}^p_{\geq \sigma}$ aims to model the mean vectors and the covariance matrices of Gaussian distributions, it is also natural to use a ground metric $c$ that is inspired by the Wasserstein distance between Gaussian distributions. Fortunately, the Wasserstein type-2 distance between Gaussian measures is known in closed form [Olkin and Pukelsheim, 1982, Givens and Shortt, 1984].

**Proposition 4.2** (Wasserstein type-2 distance between Gaussian distributions). *The Wasserstein type-2 distance between two $p$-dimensional Gaussian distributions $\mathcal{N}(\mu, \Sigma)$ and $\mathcal{N}(\widehat{\mu}, \widehat{\Sigma})$ under the Euclidean*

probability measure $\nu_y$ on the mean-covariance space of $\mathbb{R}^p \times \mathbb{S}^p_+$ such that for any measurable set $\mathcal{S} \subseteq \mathbb{R}^p$

$$\mathbb{Q}_y(X \in \mathcal{S}) = \int_{\mathbb{R}^p \times \mathbb{S}^p_+} \int_{\mathcal{S}} f(\tilde{x}|\mu, \Sigma)\, \mathrm{d}\tilde{x}\, \nu_y(\mathrm{d}\mu, \mathrm{d}\Sigma),$$

where $f(\cdot\,|\mu, \Sigma)$ is the density function of the Gaussian distribution $\mathcal{N}(\mu, \Sigma)$.

*ground metric amounts to* $\mathbb{G}(\mathcal{N}(\mu, \Sigma), \mathcal{N}(\widehat{\mu}, \widehat{\Sigma})) = \sqrt{\|\mu - \widehat{\mu}\|_2^2 + \operatorname{Tr}\left[\Sigma + \widehat{\Sigma} - 2(\widehat{\Sigma}^{\frac{1}{2}} \Sigma \widehat{\Sigma}^{\frac{1}{2}})^{\frac{1}{2}}\right]}.$

Motivated by the above result, we endow the space $\mathbb{R}^p \times \mathbb{S}^p_{\geq \sigma}$ with the cost function $c$ defined as

$$c((\mu, \Sigma), (\widehat{\mu}, \widehat{\Sigma}))$$
$$\triangleq \sqrt{\|\mu - \widehat{\mu}\|_2^2 + \operatorname{Tr}\left[\Sigma + \widehat{\Sigma} - 2(\widehat{\Sigma}^{\frac{1}{2}} \Sigma \widehat{\Sigma}^{\frac{1}{2}})^{\frac{1}{2}}\right]}.$$

It is easy to see that $c$ is non-negative, symmetric and continuous on $\mathbb{R}^p \times \mathbb{S}^p_{\geq \sigma}$ and thus $c$ is a valid ground cost for the Wasserstein distance $\mathbb{W}_c$ on $\mathbb{R}^p \times \mathbb{S}^p_{\geq \sigma}$. We should point out that the Wasserstein distance has also been heavily used to construct ambiguity sets in the context of distributionally robust machine learning Nguyen et al. [2019b], Taskesen et al. [2021], Vu et al. [2022]. Our formulation of $\mathbb{W}_c$ is related with the family of optimal transport for Gaussian mixtures, which we discuss in the next remark.

**Remark 4.3** (OT between Gaussian mixtures). *Our construction relies on representing a Gaussian mixture distribution as a discrete distribution on the mean vector and covariance matrix space. This construction is motivated by recent work on optimal transport between Gaussian mixtures in Chen et al. [2019] and Delon and Desolneux [2020]. A clear distinction is that we use $\mathbb{W}_c$ as the type-$\infty$ distance in Definition 4.1, while the existing literature focuses on type-1 and type-2 distance. As we later demonstrate in Lemma 5.3, the type-$\infty$ construction is critical for the separability of the resulting problem.*

## 5 COMPUTATION

In this section, we delineate the solution procedure to find a robust Bayesian recourse with the Wasserstein-Gaussian mixture ambiguity sets formalized in Section 4. Fix any measure $\mathbb{Q}_y \in \mathbb{B}_{\varepsilon_y}(\widehat{\mathbb{P}}^\sigma_y)$, then $X|Y = y$ follows a mixture of Gaussian under $\mathbb{Q}_y$, and we let $L(x, \mathbb{Q}_y)$ be the Gaussian mixture likelihood of a point $x$ under $\mathbb{Q}_y$. By internalizing the maximization term inside the fraction and replacing $\mathbb{Q}_y(X = x)$ by the likelihood $L(x, \mathbb{Q}_y)$, problem (2) is equivalent to

$$\min_{x \in \mathbb{X}} F(x), \quad F(x) \triangleq \frac{\gamma_0 \times \max\limits_{\mathbb{Q}_0 \in \mathbb{B}_{\varepsilon_0}(\widehat{\mathbb{P}}^\sigma_0)} L(x, \mathbb{Q}_0)}{\gamma_1 \times \min\limits_{\mathbb{Q}_1 \in \mathbb{B}_{\varepsilon_1}(\widehat{\mathbb{P}}^\sigma_1)} L(x, \mathbb{Q}_1)}.$$

In the sequence, we discuss how to evaluate the objective value $F(x)$, sketch the necessary proof and provide further insights to the likelihood evaluation problems.

### 5.1 REFORMULATIONS OF THE LIKELIHOOD EVALUATION PROBLEMS AND ROUTINES

For any $x \in \mathbb{X}$, evaluating its objective value $F(x)$ requires solving the maximization of the likelihood in the numerator

$$\max \left\{ L(x, \mathbb{Q}_0) : \mathbb{Q}_0 \in \mathbb{B}_{\varepsilon_0}(\widehat{\mathbb{P}}_0^\sigma) \right\} \qquad (3)$$

and the minimization of the likelihood in the denominator

$$\min \left\{ L(x, \mathbb{Q}_1) : \mathbb{Q}_1 \in \mathbb{B}_{\varepsilon_1}(\widehat{\mathbb{P}}_1^\sigma) \right\}. \qquad (4)$$

At this stage, it is important to relate problems (3) and (4) to the existing literature on (Bayesian) likelihood estimation/approximation. Problem (3) searches for a distribution that *maximizes* the likelihood of $x$ over the set $\mathbb{B}_{\varepsilon_0}(\widehat{\mathbb{P}}_0^\sigma)$, and it is also known in the machine learning literature as an *optimistic* likelihood Nguyen et al. [2019a, 2020]. There is, however, a clear distinction between the existing results and the results of this paper: Nguyen et al. [2019a] use a Gaussian feasible set prescribed using the Fisher-Rao distance and Nguyen et al. [2020] use a moment-based feasible set using the Kullback-Leibler type divergence; in contrast, our set $\mathbb{B}_{\varepsilon_0}(\widehat{\mathbb{P}}_0^\sigma)$ is a mixture of Gaussian feasible set prescribed using a hierarchical Wasserstein distance. The attractiveness of the existing optimistic likelihood methods lies in their computational tractability. Next, we show that our optimistic likelihood under the Wasserstein-Gaussian mixture ambiguity set also possesses this tractability.

**Theorem 5.1** (Optimistic likelihood)**.** *For each $i \in \mathcal{I}_0$, let $\alpha_i$ be the optimal value of the following two-dimensional optimization problem*

$$\min_{\substack{a \in \mathbb{R}_+, \ d_p \in [\sigma, +\infty) \\ a^2 + (d_p - \sigma)^2 \leq \varepsilon_0^2}} \log d_p + \frac{(\|x - \widehat{x}_i\|_2 - a)^2}{2d_p^2} + (p-1)\log \sigma.$$

*Then, we have*

$$\max \left\{ L(x, \mathbb{Q}_0) : \mathbb{Q}_0 \in \mathbb{B}_{\varepsilon_0}(\widehat{\mathbb{P}}_0^\sigma) \right\} = \frac{\sum_{i \in \mathcal{I}_0} \exp(-\alpha_i)}{N_0 (2\pi)^{p/2}}.$$

Theorem 5.1 asserts that we can solve problem (3) by solving $N_0$ individual subproblems, each subproblem is a two-dimensional minimization problem. Notice that the feasible set of each subproblem is relatively simple: it contains an ellipsoidal constraint and lower bounds on the variables. Hence, it is easy to devise a projection operator for this feasible set. Note that the objective function of the subproblem is non-convex.

Let us now focus our attention on problem (4): it searches for a distribution that *minimizes* the likelihood of $x$ over all candidate distributions in $\mathbb{B}_{\varepsilon_1}(\widehat{\mathbb{P}}_1^\sigma)$, and it is termed the *pessimistic* likelihood. It has been previously noticed that the pessimistic likelihood is not easy to solve due to non-convexity [Nguyen et al., 2020, Appendix A]. Surprisingly, for our Wasserstein-Gaussian mixture set, we still can obtain the reformulation below.

**Theorem 5.2** (Pessimistic likelihood)**.** *For each $i \in \mathcal{I}_1$, let $\alpha_i$ be the optimal value of the following two-dimensional*

*optimization problem*

$$\min_{\substack{a \in \mathbb{R}_+, \ d_1 \in [\sigma, +\infty) \\ a^2 + p(d_1 - \sigma)^2 \leq \varepsilon_1^2}} \left\{ -\log d_1 - \frac{(\|x - \widehat{x}_i\|_2 + a)^2}{2d_1^2} \right.$$

$$\left. -(p-1)\log \left( \sigma + \sqrt{\frac{\varepsilon_1^2 - a^2 - (d_1 - \sigma)^2}{p - 1}} \right) \right\}.$$

*Then, we have*

$$\min \left\{ L(x, \mathbb{Q}_1) : \mathbb{Q}_1 \in \mathbb{B}_{\varepsilon_1}(\widehat{\mathbb{P}}_1^\sigma) \right\} = \frac{\sum_{i \in \mathcal{I}_1} \exp(\alpha_i)}{N_1 (2\pi)^{p/2}}.$$

Theorem 5.2 asserts that the pessimistic likelihood problem (4) admits a similar decomposable structure: solving (4) is equivalent to solving $N_1$ individual subproblems, each subproblem is a two-dimensional minimization problem with a non-convex objective function. Further, the feasible set of the subproblem is also of tractable form for projection.

**Numerical routines.** Equipped with Theorems 5.1 and 5.2, we can design an iterative scheme to solve the robust Bayesian recourse problem. For any value $x \in \mathbb{X}$, we can use a projected gradient descent to solve a series of two-dimensional subproblems to evaluate the objective value $F(x)$. In Appendix C, we elaborate on the construction of the projection operator as well as the algorithm to evaluate $F(x)$. To optimize $F(x)$ to find the robust recourse, we can also apply a similar algorithm, provided that the projection onto the feasible region $\mathbb{X}$ is easy to solve.

### 5.2 SKETCH OF PROOFS

We sketch here the main steps leading to the results in Section 5.1. Because $\widehat{\mathbb{P}}_y^\sigma$ is a Gaussian mixture and we are using a type-$\infty$ Wasserstein distance to prescribe the neighborhood around the representable distribution, the likelihood evaluation problems admit a decomposable structure. This decomposability has also been exploited previously in the literature of operations management [Bertsimas et al., 2021], chance constrained programming [Xie, 2020] and fair classification [Wang et al., 2021]. In the sequel, we denote $f(x|\mu_i, \Sigma_i)$ the likelihood of $x$ under the $p$-dimensional Gaussian distribution with a mean vector $\mu_i$ and a covariance matrix $\Sigma_i$:

$$f(x|\mu_i, \Sigma_i) = \frac{\exp\left(-\frac{1}{2}(x - \mu_i)^\top \Sigma_i^{-1}(x - \mu_i)\right)}{(2\pi)^{\frac{p}{2}} \det(\Sigma_i)}.$$

The next lemma asserts that the likelihood evaluation problem can be decomposed into solving smaller subproblems, each subproblem is an optimization problem over the mean vector - covariance matrix space $\mathbb{R}^p \times \mathbb{S}_{\geq \sigma}^p$.

**Lemma 5.3** (Separability)**.** *There exists a distribution $\mathbb{Q}_0^\star$ that solves (3) and is a mixture of at most $N_0$ Gaussian components. Moreover, problem (3) is equivalent to a separable*

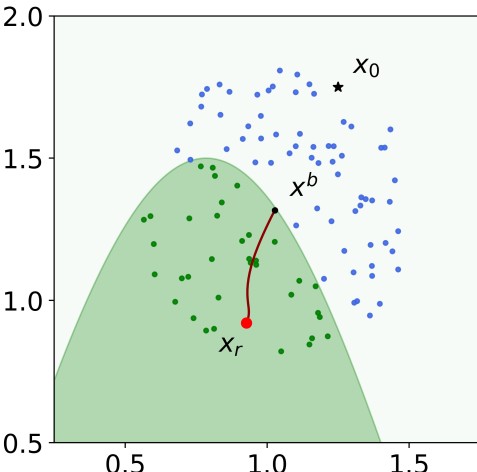

Figure 1: An example of the robust Bayesian recourse on a toy 2-dimensional instance. The star denotes the input $x_0$, and the black circle denotes the boundary point $x^b$. Green and blue circles are locally sampled data with favorable and unfavorable predicted values, respectively. The red circle denotes the robust Bayesian recourse, and the curved line denotes the continuum of intermediate solutions of the gradient descent algorithm. The robust Bayesian recourse moves to the interior of the favorable region (green), and thus is more likely to be valid subject to model shifts.

*problem of the form*

$$\max \left\{ L(x, \mathbb{Q}_0) : \mathbb{Q}_0 \in \mathbb{B}_{\varepsilon_0}(\widehat{\mathbb{P}}_0^\sigma) \right\}$$
$$= \begin{cases} \max & \frac{1}{N_0} \sum_{i \in \mathcal{I}_0} f(x|\mu_i, \Sigma_i) \\ \text{s.t.} & (\mu_i, \Sigma_i) \in \mathbb{R}^p \times \mathbb{S}^p_{\geq \sigma} \\ & c((\mu_i, \Sigma_i), (\widehat{x}_i, \sigma^2 \bar{I})) \leq \varepsilon_0 \quad \forall i \in \mathcal{I}_0. \end{cases}$$

*An analogous result holds for problem (4) with the corresponding subscript $y = 1$.*

Lemma 5.3 leverages the essential supremum in the definition of the type-$\infty$ Wasserstein distance in Definition 4.1 to separate the problem into subproblem for each component. This separability is *not* obtainable under other types of the Wasserstein distance. It is important to bear in mind that each subproblem is still not easy: the objective function is neither convex nor concave in $\Sigma_i$. Further, we also need to evaluate both the maximization and the minimization counterparts, and tractability is difficult to be established simultaneously in both directions. Despite these difficulties, we can show that each subproblem, which is originally on the $\mathbb{R}^p \times \mathbb{S}^p_{\geq \sigma}$ space, can be reduced to a 2-dimensional subproblem. This is in fact a significant reduction of dimensionality, and this reduction does not depend on the dimension $p$. First, we provide the reformulation for the maximization counterpart.

**Proposition 5.4** (Maximization subproblem). *Fix any index*

$i \in \mathcal{I}_0$. *For any $\widehat{x}_i \in \mathbb{R}^p$, $x \in \mathbb{R}^p$ and $\varepsilon_0 \in \mathbb{R}_+$, we have*

$$\frac{\exp(-\alpha_i)}{(2\pi)^{p/2}} = \begin{cases} \max & f(x|\mu_i, \Sigma_i) \\ \text{s.t.} & (\mu_i, \Sigma_i) \in \mathbb{R}^p \times \mathbb{S}^p_{\geq \sigma} \\ & c((\mu_i, \Sigma_i), (\widehat{x}_i, \sigma^2 \bar{I})) \leq \varepsilon_0, \end{cases}$$

*where $\alpha_i$ is the optimal value of the two-dimensional optimization problem*

$$\min_{\substack{a \in \mathbb{R}_+, \ d_p \in [\sigma, +\infty) \\ a^2 + (d_p - \sigma)^2 \leq \varepsilon_0^2}} \log d_p + \frac{(\|x - \widehat{x}_i\|_2 - a)^2}{2 d_p^2} + (p-1) \log \sigma.$$

The two auxiliary variables $a$ and $d_p$ have a specific meaning which can be explained as follows. Let $(\mu_i^\star, \Sigma_i^\star)$ be the optimal solution of the original maximization problem over $(\mu_i, \Sigma_i)$, and let $(a^\star, d_p^\star)$ be the optimal solution of the reduced problem over $(a, d_p)$. We then have $\|\mu_i^\star - \widehat{x}_i\|_2 = a^\star$ and $d_p^\star$ coincides with the *largest* eigenvalues of $\Sigma_i^\star$. Next, we expose the reformulation for the minimization problem.

**Proposition 5.5** (Minimization subproblem). *Fix any index $i \in \mathcal{I}_1$. For any $\widehat{x}_i \in \mathbb{R}^p$, $x \in \mathbb{R}^p$ and $\varepsilon_1 \in \mathbb{R}_+$, we have*

$$\frac{\exp(\alpha_i)}{(2\pi)^{p/2}} = \begin{cases} \min & f(x|\mu_i, \Sigma_i) \\ \text{s.t.} & (\mu_i, \Sigma_i) \in \mathbb{R}^p \times \mathbb{S}^p_{\geq \sigma} \\ & c((\mu_i, \Sigma_i), (\widehat{x}_i, \sigma^2 \bar{I})) \leq \varepsilon_1, \end{cases}$$

*where $\alpha_i$ is the optimal value of the two-dimensional optimization problem*

$$\min_{\substack{a \in \mathbb{R}_+, \ d_1 \in [\sigma, +\infty) \\ a^2 + p(d_1 - \sigma)^2 \leq \varepsilon_1^2}} \left\{ -\log d_1 - \frac{(\|x - \widehat{x}_i\|_2 + a)^2}{2 d_1^2} \right.$$
$$\left. -(p-1) \log \left( \sigma + \sqrt{\frac{\varepsilon^2 - a^2 - (d_1 - \sigma)^2}{p-1}} \right) \right\}.$$

There is a similar relationship between $(\mu_i^\star, \Sigma_i^\star)$ which solves the original minimization problem and $(a^\star, d_1^\star)$ which solves the reduced problem: we have $\|\mu_i^\star - \widehat{x}_i\|_2 = a^\star$ and $d_1^\star$ coincides with the *smallest* eigenvalues of $\Sigma_i^\star$.

The above discussion reveals that we can fully reconstruct the distribution $\mathbb{Q}_0^\star$ that solves (3) and $\mathbb{Q}_1^\star$ that solves (4) from the solutions of the reduced subproblems, we provide this reconstruction in Appendix D.

# 6 NUMERICAL EXPERIMENT

We evaluate in this section the robustness to model shifts of different recourses, together with the trade-off against the cost of adopting the recourse's recommendation. We compare our proposed robust Bayesian recourse method, namely RBR, against the counterfactual explanation of Wachter [Wachter et al., 2017] and against the robust recourse generated by ROAR [Upadhyay et al., 2021] using

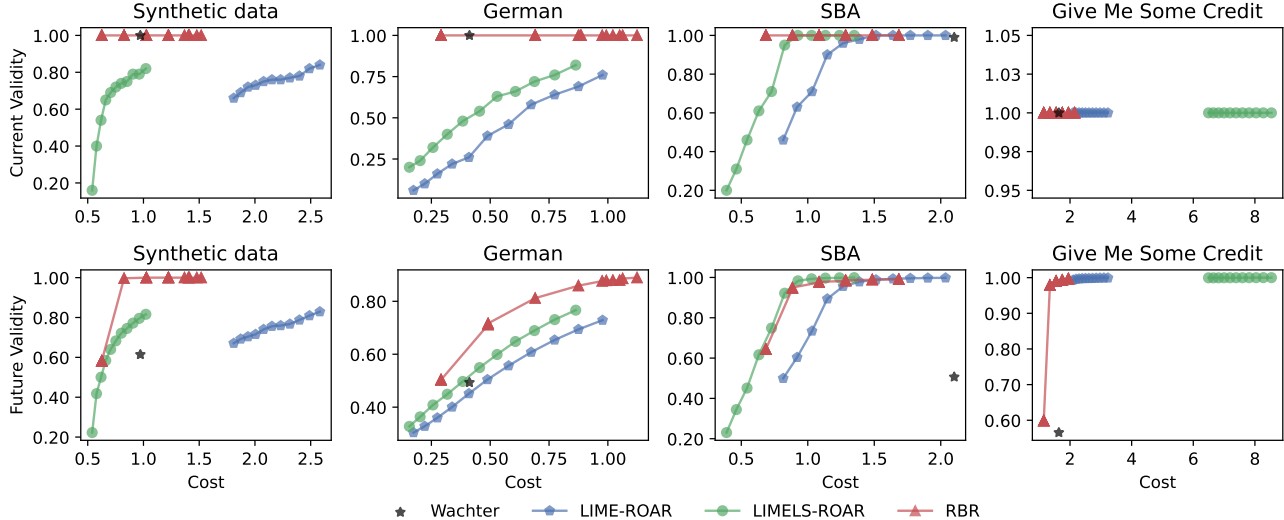

Figure 2: Pareto frontiers of the cost-validity trade-off with the MLP classifier, on synthetic, German Credit, Small Business Administration, and Give Me Some Credit datasets.

either LIME [Ribeiro et al., 2016] and LIMELS [Laugel et al., 2018] as a surrogate model[3].

## 6.1 EXPERIMENTAL SETUP

**Datasets.** We examine the recourse generators on both a synthetic dataset and the real-world datasets: *German Credit* [Dua and Graff, 2017, Groemping, 2019], *Small Bussiness Administration (SBA)* [Li et al., 2018], and *Give Me Some Credit (GMC)*. Each dataset contains two sets of data: $D_1$ and $D_2$. The former is the current data which is used to train current classifier to generate recourses. The latter represents the possible data arriving in the future.

For each dataset, we use 80% of the instances in the current data $D_1$ to train the underlying predictive model and fix this classifier to construct recourses for the remaining 20% of the instances. The future data $D_2$ will be used to train future classifiers, which are for evaluation only.

**Classifier.** We use a three-layer MLP with 20, 50 and 20 nodes, respectively with a ReLU activation in each consecutive layer. The sigmoid function is used in the last layer to produce predictive probabilities. The performance of the MLP classifier is reported in Table 1.

**Sampling procedure.** We employ the sampling scheme described in Section 2. We choose the number of counterfactuals $K = 1000$ and sample 200 synthetic samples uniformly with a sampling radius $r_p = 0.2$.

---

[3]While LIME samples synthetic data *globally* and train a weighted ridge regression, LIMELS generates the local surrogate model by training a (unweighted) ridge regression on the data sampled *locally* near by the closest counterfactual of the input instance (similar to the sampling procedure described in Section 2).

**Metrics.** To measure the ease of adopting a recourse, we use the $\ell_1$-distance as the cost function $\varphi$ on the covariate space $\mathcal{X}$, this choice is similar to Ustun et al. [2019] and Upadhyay et al. [2021]. We define the *current validity* as the validity of the recourses with respect to the current classifier $\mathcal{C}$. To evaluate the robustness of recourses to the changes in model's parameters, we sample 20% of the instances in the data set $D_2$ as the arrival data. We then re-train the classifier with the old data (80% of $D_1$) coupled with this arrival data to simulate the future classifiers $\tilde{\mathcal{C}}$. We repeat this procedure 100 times to obtain 100 future classifiers and report the *future validity* of a recourse as the fraction of the future classifiers with respect to which the recourse is valid.

## 6.2 EXPERIMENTAL DETAILS

We use both synthetic and real-world datasets.

**Synthetic dataset.** We synthesize the 2-dimensional data by sampling 1000 instances uniformly in a rectangle $[-2, 4] \times [-2, 7]$. For each sample, we label using the function $f(x) = 1$ if $x_2 \geq 1 + x_1 + 2x_1^2 + x_1^3 - x_1^4 + \varepsilon$, and $f(x) = $ otherwise, where $\varepsilon$ is a random noise. We set $\varepsilon = 0$ when generating the present set $D_1$ and $\varepsilon \sim \mathcal{N}(0, 1)$ for the future set $D_2$.

**Real-world datasets.** Three real-world datasets are used.

- *German Credit* [Dua and Graff, 2017]. The dataset contains the information (e.g. age, gender, financial status,...) of 1000 customers who took a loan from a bank. The classification task is to determine the risk (good or bad) of an individual. There is another version of this dataset regarding to corrections of coding error [Groemping, 2019]. We use the corrected version of

| Dataset | Present data $D_1$ | | Shift data $D_2$ | |
|---|---|---|---|---|
| | *Accuracy* | *AUC* | *Accuracy* | *AUC* |
| Synthetic data | $0.99 \pm 0.00$ | $1.00 \pm 0.00$ | $0.94 \pm 0.01$ | $0.99 \pm 0.01$ |
| German Credit | $0.67 \pm 0.02$ | $0.60 \pm 0.03$ | $0.66 \pm 0.23$ | $0.60 \pm 0.04$ |
| SBA | $0.96 \pm 0.00$ | $0.99 \pm 0.00$ | $0.98 \pm 0.01$ | $0.96 \pm 0.01$ |
| GMC | $0.94 \pm 0.00$ | $0.84 \pm 0.00$ | $0.94 \pm 0.00$ | $0.84 \pm 0.00$ |

Table 1: Accuracy and AUC results of the MLP classifier on the synthetic and real-world datasets.

this dataset as a shifted data to capture correction shift. The features we used in this dataset include 'duration', 'amount', 'personal_status_sex', and 'age'.

- *Small Bussiness Administration (SBA)* [Li et al., 2018]. This dataset includes 2102 observations of small business loan approvals from 1987 to 2014. We divide it into two datasets (one is instances from 1989 - 2006 and one is instances from 2006 - 2014) to capture temporal shift. We use the following features: 'Term', 'NoEmp', 'CreateJob', 'RetainedJob', 'UrbanRural', 'ChgOffPrinGr', 'GrAppv', 'SBA_Appv', 'New', 'RealEstate', 'Portion', 'Recession'.

- *Give Me Some Credit (GMC)*[4]. This dataset is used to predict if a person would experience financial distress in the next two years. Given 150000 entries from the available dataset, we randomly shuffle and partition the data equally into the current set $D_1$ and the shifted set $D_2$. Each entry contains 10 features: 'RevolvingUtilizationOfUnsecuredLines', 'age', 'NumberOfTime30-59DaysPastDueNotWorse', 'DebtRatio', 'MonthlyIncome', 'NumberOfOpenCreditLinesAndLoans', 'NumberOfTimes90DaysLate', 'NumberRealEstateLoansOrLines', 'NumberOfTime60-89DaysPastDueNotWorse', 'NumberOfDependents'.

### 6.3   COST-VALIDITY TRADE-OFF

We obtain the Pareto front for the trade-off between the cost of adopting recourses produced by RBR and their validity by varying the ambiguity sizes $\varepsilon_1$ and $\varepsilon_0$, along the maximum recourse cost $\delta$, with $\delta = \|x_0 - x_b\|_1 + \delta_+$. Particularly, we consider $\sigma = 1.0$, $\varepsilon_0, \varepsilon_1 \in \{0.5k \mid k = 0, \ldots, 2\}$, and $\delta_+ \in \{0.2l \mid l = 0, \ldots, 5\}$. The frontiers for ROAR-based methods are obtained by varying $\delta_{\max} \in \{0.02m \mid m = 0, \ldots, 10\}$, where $\delta_{\max}$ is the tuning parameter of ROAR. As shown in Figure 2, increasing $\varepsilon_1$ and $\delta_+$ generally increase the future validity of recourses yielded by RBR at the sacrifice of the cost, while sustaining the current validity. Yet, the frontiers obtained by RBR either dominate or comparable to other frontiers of Wachter, LIME-ROAR, and LIMELS-ROAR.

**Conclusions.** In this work, we proposed the robust Bayesian recourse which aims to be effective at reversing algorith-

mic outcome under potential model shifts. It is a model-agnostic approach that does not require approximating the nonlinear classifier by a linear surrogate. Instead, the robust Bayesian recourse minimizes directly the worst-case posterior probability odds ratio subject to the cost constraint bound. The robustness is designed with respect to the Wasserstein-Gaussian mixture ambiguity sets of the conditional distributions, in which the neighborhood is prescribed using an optimal transport (type-$\infty$ Wasserstein) distance. We showed that the min-max recourse problem can be optimized using a gradient descent algorithm, which exploits separability and dimensionality reduction when evaluating the objective value. Our experiments on synthetic and real-world datasets demonstrate that the robust Bayesian recourse is more robust at a lower cost than other baselines.

While this paper focus on algorithmic transparency, we note that transparency may lead to the tension between transparency and gaming-the-system behaviors: the greater transparent be the decision process the more opportunity for exploitative manipulations [Yan and Zhang, 2022]. We envision that robustness techniques may alleviate these gaming behaviors and may lead to more trustworthy guarantees of (machine learning) algorithms.

**Acknowledgments.** Man-Chung Yue is supported by the HKRGC under the General Research Fund project 15305321.

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
