# OpenReview forum: "Robust Bayesian Recourse"
_auai.org/UAI/2022/Conference — UAI 2022 Poster_

### Official Review · Reviewer_y7L3 · 2022-04-07

**Q2(1) Originality/Novelty:** 4
**Q2(2) Significance/Impact:** 3
**Q2(3) Correctness/Technical Quality:** 3
**Q2(6) Clarity Of Writing:** 4
**Q6 Overall Score:** 7
**Q8 Confidence In Your Score:** 2

**Q1 Summary And Contributions:**

The paper proposes a Bayesian resource -- a model-agnostic method that aims to reverse algorithmic outcomes under potential model shifts. In other words, the method provides a way to answer the question "what action can be taken in order to receive an alternative algorithmic outcome". Bayesian resource minimizes the odds ratio between the posterior probability of negative and positive predicted outcomes and aims to be robust to changes in the parameters defining the classifier.

**Q10 Ethical Concerns (Optional):**

No.

**Q2 Assessment Of The Paper:**

More detailed information regarding each of these aspects is given below:

**Q2(4) Quality Of Experiments (Optional):**

3: Good: The experimental evaluation is adequate, and the results convincingly support the main claims.

**Q2(5) Reproducibility:**

4: Excellent: Key resources (e.g., proofs, code, data) are available and key details (e.g., proof sketches, experimental setup) are comprehensively described for competent researchers to confidently and easily reproduce the main results.

**Q3 Main Strengths:**

The paper is very well-written and clearly motivated. The topic is quite specialized, but the problem is introduced in a very friendly way while not lacking in technical details.

**Q4 Main Weakness:**

None that I can name.

**Q5 Detailed Comments To The Authors:**

While not an expert on this particular topic, I found the paper easy enough to follow and genuinely enjoyed reading it.

**Q7 Justification For Your Score:**

The paper is very clearly written and very well motivated. The reproducibility of the results is supported by extensive details and submitted code.

**Q9 Complying With Reviewing Instructions:**

1: Yes.

---

### Official Review · Reviewer_oFHC · 2022-04-12

**Q2(1) Originality/Novelty:** 3
**Q2(2) Significance/Impact:** 3
**Q2(3) Correctness/Technical Quality:** 3
**Q2(6) Clarity Of Writing:** 3
**Q6 Overall Score:** 6
**Q8 Confidence In Your Score:** 3

**Q1 Summary And Contributions:**

The authors proposed a method to provide a model agnostic algorithmic recourse that is robust to distributional shifts. The main contributions are: 1) unlike existing methods of generating recourse, the presented formulation does not depend on a linear surrogate of the original non-linear model. Instead a Bayesian approach is proposed to generate recourse. 2) to improve the robustness of the recourse, a method similar to adversarial perturbation of the empirical conditional distribution is used


**Q2 Assessment Of The Paper:**

More detailed information regarding each of these aspects is given below:

**Q2(4) Quality Of Experiments (Optional):**

3: Good: The experimental evaluation is adequate, and the results convincingly support the main claims.

**Q2(5) Reproducibility:**

3: Good: Key resources (e.g., proofs, code, data) are available and key details (e.g., proofs, experimental setup) are sufficiently well-described for competent researchers to confidently reproduce the main results.

**Q3 Main Strengths:**

1. Formulation of recourse as a Bayesian optimization problem.
2. This approach considers the original non-linear classifier in finding robust recourse, unlike prior work that considers linear approximation of the non-linear model.
3. Reformulating the inner maximization as two subproblems that can be solved using projected gradient descent.


**Q4 Main Weakness:**

The experimental section requires clarity and improvement, and an analysis of the results is missing.

**Q5 Detailed Comments To The Authors:**

Although the main paper is well written, it is difficult to follow the section on proofs; it would be good if you could restate the proposition and provide corresponding proof.

**Q7 Justification For Your Score:**

This paper presented a new approach for finding robust recourse and showed its efficiency by good results. Apart from a few minor issues regarding the writing style of the experimental section and proofs, this paper has no significant issues.

**Q9 Complying With Reviewing Instructions:**

1: Yes.

---

### Official Review · Reviewer_SCaQ · 2022-04-12

**Q2(1) Originality/Novelty:** 2
**Q2(2) Significance/Impact:** 2
**Q2(3) Correctness/Technical Quality:** 2
**Q2(6) Clarity Of Writing:** 2
**Q6 Overall Score:** 3
**Q8 Confidence In Your Score:** 2

**Q1 Summary And Contributions:**

A method is proposed for recourse: recommending an alternative, nearby value of the covariates for which a classifier would assign a positive outcome instead of the received negative outcome. The method proposed here is based on Bayesian posterior odds, and is constructed to be robust to changes in the classifier.

**Q2 Assessment Of The Paper:**

More detailed information regarding each of these aspects is given below:

**Q2(5) Reproducibility:**

2: Fair: Key resources (e.g., proofs, code, data) are unavailable but key details (e.g., proof sketches, experimental setup) are sufficiently well-described for an expert to confidently reproduce the main results.

**Q3 Main Strengths:**

Robust recourse can be expected to be important in applications.

**Q4 Main Weakness:**

Section 2 gave me the feeling the paper was trying to sell the method as something it is not.

**Q5 Detailed Comments To The Authors:**

Definition 2.1 defines a Bayesian recourse $x_\text{Bayes}$ as a point $x$ in some region minimizing $P(\hat{Y}=0 | X=x) / P(\hat{Y}=1 | X=x)$. This is equivalent to minimizing $P(\hat{Y}=0 | X=x)$. The relevance of the links drawn to Bayesian statistics in the next paragraph isn't clear to me because of this, and also because neither of the two variables involved is a model parameter.

In the paragraph below that, "we can argue that $P(\hat{Y}=0 | X=x_\text{Bayes})$ tends to be low" -- actually, you can say more strongly that $x_\text{Bayes}$ minimizes this quantity.

Other comments:
* Abstract: "probability odds" - these terms contradict each other
* The first sentence of the introduction reads like this paper will be about planning. I suggest rewriting this part of the introduction.
* page 2 top right: "our method ... employ" -> "... employs"; "the followings" -> "the following"
* Sampling scheme, bullet 3: min should be argmin


**Q7 Justification For Your Score:**

Unfortunately, after getting a negative impression early on and not being familiar with other recourse literature, I could not justify spending more time to review this paper.

**Q9 Complying With Reviewing Instructions:**

1: Yes.

---

### Official Review · Reviewer_GeQs · 2022-04-13

**Q2(1) Originality/Novelty:** 2
**Q2(2) Significance/Impact:** 2
**Q2(3) Correctness/Technical Quality:** 2
**Q2(6) Clarity Of Writing:** 3
**Q6 Overall Score:** 3
**Q8 Confidence In Your Score:** 2

**Q1 Summary And Contributions:**

The paper proposes a method for algorithmic recourse to recommend overturning a machine learning decision. The idea is to minimize the posterior probability odds ratio, which is model-agnostic. The method is extended to a robust version that considers possible perturbations of the data. Compared to the related work, the method does not require a linear approximation step.

**Q2 Assessment Of The Paper:**

More detailed information regarding each of these aspects is given below:

**Q2(4) Quality Of Experiments (Optional):**

2: Fair: The experimental evaluation is weak: important baselines are missing, or the results do not adequately support the main claims.

**Q2(5) Reproducibility:**

3: Good: Key resources (e.g., proofs, code, data) are available and key details (e.g., proofs, experimental setup) are sufficiently well-described for competent researchers to confidently reproduce the main results.

**Q3 Main Strengths:**

1) The paper tackles an interesting problem, and proposes a model-agnostic approach.
2) The approach removes the need for an intermediate linear model.


**Q4 Main Weakness:**

While the paper tackles an interesting problem, I have some concerns:
1) I am not sure how the method answers counterfactual questions without a causal model?
2) The robust bayesian recourse does not cover all possible recourses that would change the decision? I am not sure how it will be used to turn over a decision.
3) The experiments section show good results on synthetic data only, and does not appear to be statistically significant.

**Q5 Detailed Comments To The Authors:**

1) Figure 1 does not appear to be referenced in the text.

**Q7 Justification For Your Score:**

The method is not clear to me. In particular, how are counterfactual questions answered without having a causal graphical model? How do we control for spurious correlations? The experiments section would need to be further improved.

**Q9 Complying With Reviewing Instructions:**

1: Yes.

---

### Decision · Program_Chairs · 2022-05-15

**Decision:**

Accept (Poster)

**Comment:**

Meta Review: The reviews of this paper were mixed. Two of the reviewers gave it solid "accept" scores, while two reviewers gave it a very low score. After the author feedback and discussion, one of those reviewers said that the response seemed reasonable (though did not update his score). The second reviewer did not believe the paper was "Bayesian," even after the feedback; the rest of that review was not very informative. I then checked the paper myself; the statistical underpinnings of the work are posterior (log) odds, so I believe it is, in fact, "Bayesian." The reviewer also had a low confidence score. Thus, I tend to discount that particular review somewhat.

For reference, the most confident reviewer recommended a "weak accept".

Aside from the "Bayesian" considerations, the reviewers agreed that the paper was clearly written, that the problem is generally important, and that avoiding the linear approximation common in similar approaches is an interesting, novel approach. Considering that, I tend to recommend accepting the paper due to its novelty, while acknowledging the writing could likely be improved in a future version.

Pros
* Important problem
* Model agnostic
* Removes linear approximations used in other methods
* Efficient optimization

Cons
* Does not discuss counterfactuals
* Experimental result discussion is very limited
* Proofs hard to follow